# Ability of Different Growth Indicators to Detect Salt Tolerance of Advanced Spring Wheat Lines Grown in Real Field Conditions

**DOI:** 10.3390/plants13060882

**Published:** 2024-03-19

**Authors:** Muhammad Mubushar, Salah El-Hendawy, Yaser Hassan Dewir, Nasser Al-Suhaibani

**Affiliations:** Department of Plant Production, College of Food and Agriculture Sciences, King Saud University, Riyadh 11451, Saudi Arabia

**Keywords:** grain yield, green leaf area, leaf area duration, net assimilation rate, plant dry weight, relative growth rate, specific leaf weight

## Abstract

Plant growth indicators (GIs) are important for evaluating how different genotypes respond to normal and stress conditions separately. They consider both the morphological and physiological components of plants between two successive growth stages. Despite their significance, GIs are not commonly used as screening criteria for detecting salt tolerance of genotypes. In this study, 36 recombinant inbred lines (RILs) along with four genotypes differing in their salt tolerance were grown under normal and 150 mM NaCl in a two-year field trial. The performance and salt tolerance of these germplasms were assessed through various GIs. The analysis of variance showed highly significant variation between salinity levels, genotypes, and their interaction for all GIs and other traits in each year and combined data for two years, with a few exceptions. All traits and GIs were significantly reduced by salinity stress, except for relative growth rate (RGR), net assimilation rate (NAR), and specific leaf weight (SLW), which increased under salinity conditions. Traits and GIs were more correlated with each other under salinity than under normal conditions. Principal component analysis organized traits and GIs into three main groups under both conditions, with RGR, NAR, and specific leaf area (SLA) closely associated with grain yield (GY) and harvest index, while leaf area duration (LAD) was closely associated with green leaf area (GLA), plant dry weight (PDW), and leaf area index (LAI). A hierarchical clustering heatmap based on GIs and traits organized germplasms into three and four groups under normal and salinity conditions, respectively. Based on the values of traits and GIs for each group, the germplasms varied from high- to low-performing groups under normal conditions and from salt-tolerant to salt-sensitive groups under salinity conditions. RGR, NAR, and LAD were important factors determining genotypic variation in GY of high- and low-performing groups, while all GIs, except leaf area duration (LAR), were major factors describing genotypic variation in GY of salt-tolerant and salt-sensitive groups. In conclusion, different GIs that reveal the relationship between the morphological and physiological components of genotypes could serve as valuable selection criteria for evaluating the performance of genotypes under normal conditions and their salt tolerance under salinity stress conditions.

## 1. Introduction

The world is currently facing major salinity issues, which are causing a significant decrease in the growth and production of essential food crops, especially in arid and semiarid regions. Several studies have forecasted that around 50% of productive lands will become unproductive by 2050 due to salinity, raising serious concerns about global food security in the near future [1]. Salinity problems affect more than 7% of the world’s total land area, 20% of arable land, and 33% of irrigated agricultural lands. Additionally, salinity issues in the agricultural sector result in an annual loss of more than USD 12 billion due to a 30–50% decrease in the production of essential food crops in salinity-affected areas [2,3,4]. Addressing the salinity issue in agriculture is crucial in achieving a 50–70% increase in grain yields for key food crops such as wheat, rice, and maize. This increase is essential to meet the needs of the projected 10 billion people by 2050 [5].

Wheat is the dominant crop on 2.1 million km^2^ of arable land, but its productivity is hindered by several environmental stresses, especially salinity stress, which is common in arid and semiarid regions [6,7]. Bread wheat is a moderately salt-tolerant crop, but high salinity levels can reduce its yield by more than 50% [3,8]. Therefore, cultivating wheat in saline conditions is challenging without implementing practices to mitigate the negative effects of salinity stress on its growth and yield. Agronomic practices such as applying mineral gypsum, organic amendments, and effective drainage schemes can help alleviate these negative impacts [9,10]. However, these practices can be costly and require expertise, particularly when implemented on a large scale. Previous studies have concurred that providing farmers with salt-tolerant genotypes is a feasible and effective strategy for addressing salinity in agriculture and achieving reasonable yields [8,11,12,13]. Despite the research community’s great efforts to enhance the salinity tolerance of wheat genotypes, the number of salt-tolerant genotypes worldwide remains very limited. Based on published studies, the reasons behind this include (1) the salt tolerance of genotypes is usually assessed under tightly controlled conditions with few experiments carried out under real environmental conditions [3,8,14]; (2) the salt tolerance of genotypes is usually assessed based on one or two developmental growth stages, particularly the early ones (germination and seedling), despite the fact that the ranking of genotypes for their salt tolerance often varies across different growth stages [15]; (3) evaluating the salt tolerance of genotypes often requires finding suitable screening criteria and accurate evaluation methods that have the potential to make the assessment of salt tolerance among different genotypes more precise and efficient, particularly across multiple traits [16,17,18]; (4) the lack of specific indicators that accurately reflect both the physiological and agronomic behavior of genotypes [3,19]; and (5) the use of different growth indicators (GIs) as screening criteria is not common. These GIs can provide insight into the strong correlation between physiological tolerance and agronomic traits, as well as the close relationship between source and sink organs. They also shed light on how biomass is allocated between different plant organs and the duration of active photosynthesis [20,21,22].

Salinity stress significantly impacts plant growth and production through osmotic stress, ion toxicity, and essential nutrient deficiencies [11,23]. This leads to the excessive buildup of toxic ions in the leaf blade, restricted water and essential nutrient uptake, and significant alterations in physiological and biochemical processes at various plant levels. These changes include inhibiting cell division and elongation, accelerating cell death and leaf senescence, increasing leaf degradation, reducing leaf growth, inhibiting photosystem II (PSII) activity, decreasing stomatal conductance, destroying photosynthetic pigments, reducing carbohydrate supply to young leaves and grains, limiting photosynthetic rate, biomass accumulation, and source-sink activity, and altering plant water status [3,11,24,25,26,27]. In the end, all of the aforementioned negative effects of salinity stress have a detrimental impact on the various agro-morphological traits of wheat plants [15,24,28,29]. Thus, different agro-morphological traits related to salt tolerance can be used as screening criteria to improve the adaptation of wheat genotypes to salinity stress. These traits can be assessed at different growth stages, and therefore, understanding their contribution to salt tolerance mechanisms at specific developmental stages is essential in improving breeding techniques and developing salt-tolerant genotypes. This will make the assessment of salt tolerance among genotypes more effective. Generally, to enhance the precision and efficiency of evaluating salt tolerance among genotypes, it is essential to simultaneously assess multiple traits at various growth stages [30,31]. However, this approach can be costly and time-consuming, particularly when dealing with a large number of genotypes.

In general, changes in plant performance between two successive growth stages are closely related to changes in several morphological and physiological components under normal or stress conditions. Previous studies have reported that various GIs, including relative growth rate (RGR), net assimilation rate (NAR), leaf area duration (LAD), leaf area ratio (LAR), specific leaf area (SLA), leaf weight ratio (LWR), and specific leaf weight (SLW), can be used to comprehensively understand how different genotypes respond to various environmental conditions at the morphological and/or physiological levels. Therefore, they could provide a suitable approach to crop improvement programs in both normal and stress conditions [21,32,33,34]. For instance, RGR, the most commonly used growth analysis component, is used to assess how dry matter is accumulated and distributed between different parts of the plant, and to determine if differences in growth performance among different genotypes are attributed to changes in morphological and/or physiological components. This is because RGR is a combination of morphological components (LAR) and physiological components (NAR) of growth [35]. NAR is a growth metric that quantifies the increase in dry biomass per unit leaf area, and is primarily influenced by the balance between photosynthesis and respiration rates of the entire plant. On the other hand, LAR represents the ratio between the photosynthetic organs (leaf area) and the photosynthetic product (total dry mass), making it a product of SLA (the ratio of surface leaf area to leaf biomass) and LWR (the ratio of leaf biomass to total plant biomass) [35]. Cramer et al. [36] found that salinity stress at the early growth stage significantly affected RGR, LAR, SLA, and LWR, and these factors were correlated with differences in salt tolerance between maize hybrids. However, NAR was not significantly affected by salinity stress and was not associated with differences in salt tolerance between hybrids. This suggests that the salt tolerance of hybrids is linked to reduced growth (dry matter accumulation) and leaf expansion, rather than the photosynthetic rate. In contrast, other studies have shown that differences in salt tolerance between wheat genotypes were associated with both RGR and NAR, indicating that both leaf expansion and photosynthetic rate play important roles in the salt tolerance of wheat genotypes [23]. Hence, different GIs, which can be calculated using simple mathematical equations, can help us understand the variations in salt tolerance among different genotypes. This makes them useful screening criteria for evaluating the salt tolerance of wheat genotypes.

Despite the importance of various GIs in understanding the complex relationships between morphological and physiological components and their response to various environmental growth conditions, there is a lack of research on using GIs as screening criteria to evaluate the growth and production performance of genotypes under normal conditions and their salt tolerance under salinity conditions. Additionally, there is also a lack of studies on using GIs to assess whether genotypic variation in production under normal conditions and salt tolerance under salinity conditions is linked to differences in morphological and/or physiological characteristics. Therefore, this study aimed to assess the effectiveness of different GIs in identifying genotypes with good performance under normal conditions and salt tolerance under salinity stress. Additionally, the study aimed to determine if genotypic variation in salt tolerance is associated with differences in morphological and/or physiological characteristics.

## 2. Results

### 2.1. Effects of Different Treatments and Their Interaction on Agro-morphological Traits and Growth Indicators

The F-values from the analysis of variance (ANOVA) show that both salinity levels (SL) and genotypes (G) had significant effects on all agro-morphological traits (traits) and growth indicators (GIs) in each year (Y) and in the combined analysis of data for both years, except for harvest index (HI) and LAR in the first year, which were not affected by SL and G, respectively (Table 1). The SL × G interaction had significant effects on all traits and GIs, except for LAR in the first year and green leaf area measured at 90 days after sowing (GLA-90), LAD, and SLA in the second year. The Y had significant effects on all traits and GIs except for plant dry weight measured at 90 days after sowing (PDW-90) and HI. Three out of eight traits (GLA-90, PDW-75, and LAI-90) and GIs related to morphological components (LAD, SLA, and SLW) were significantly affected by the SL × Y interaction. The G × Y interaction had significant effects on all traits except for GLA-90 and PDW-70. Additionally, two GIs related to physiological components (RGR, NAR) and one related to morphological components (LAD) were also significantly affected by this interaction. The G × SL × Y interaction had significant effects on all traits except for PDW-75, GY, and HI. Additionally, RGR, NAR, and LAR, which are GIs related to physiological and morphological components, were also significantly affected by this three-way interaction (Table 1).

### 2.2. Variation of Agro-Morphological Traits and Growth Indicators among Genotypes under Normal and Salinity Conditions

Figure 1 (box plot) illustrates the variability of traits and GIs among genotypes under both normal and salinity conditions. Meanwhile, Table 2 provides a summary of the descriptive statistics for traits and GIs in response to both conditions across 36 genotypes. According to Figure 1, most of the traits and GIs showed high variability among genotypes under both conditions. According to Table 2, the maximum values for the traits and GIs were approximately one to two times greater than the minimum values, regardless of whether under normal or salinity conditions. The descriptive statistics values, including range, 1st quartile, and 3rd quartile, also indicated a wide range of variation in traits and GIs among genotypes in both conditions. Additionally, the values of traits and GIs were notably reduced under high salinity levels (150 mM NaCl) compared to the normal treatment, with the exception of GIs related to physiological components (RGR and NAR) and SLW, which exhibited a significant increase under salinity stress. For instance, when compared to the control treatment, salinity stress resulted in a decrease of several traits by 7.1% to 49.3% and GIs related to morphological components (LAD, LAR, and SLA) by 14.0% to 46.5% (Table 2).

**Table 1 plants-13-00882-t001:** F-values of analysis of variance (ANOVA) for the effects of salinity level (SL), genotype (G), and their interaction on different agro-morphological traits measured at 75 and 90 days after sowing and growth indicators for each year (Y) and combined two years. The units of traits and GIs are mentioned in Figure 1.

Source	First Year			Second Year			Combined Two Years						
	SL	G	G × SL	SL	G	G × SL	Y	SL	SL × Y	G	G × Y	G × SL	G × SL × Y
df	1	35	35	1	35	35	1	1	1	35	35	35	35
Green leaf area (GLA-75)	391.35 **	10.83 ***	4.69 ***	399.72 **	4.24 ***	2.49 ***	769.70 **	788.19 ***	3.99 ^ns^	11.17 ***	2.29 ***	4.57 ***	2.07 ***
Green leaf area (GLA-90)	3784.02 ***	7.40 ***	3.74 ***	884.86 **	3.07 ***	1.31 ^ns^	28.81 *	3090.96 ***	11.47 *	7.45 ***	1.45 ^ns^	2.23 ***	1.94 **
Plant dry weight (PDW-75)	2217.97 ***	6.13 ***	2.80 ***	762.07 **	4.35 ***	2.02 **	136.76 **	2423.42 ***	12.00 *	9.13 ***	1.20 ^ns^	3.89 ***	0.88 ^ns^
Plant dry weight (PDW-90)	12,883.85 ***	8.99 ***	5.60 ***	573.37 **	7.69 ***	4.82 ***	1.19 ^ns^	2351.10 ***	3.19 ^ns^	14.66 ***	2.00 **	8.93 ***	1.49 *
Leaf area index (LAI-75)	1183.05 ***	7.37 ***	2.72 ***	8918.05 ***	2.78 ***	1.68 *	4851.17 ***	4228.80 ***	0.26 ^ns^	6.52 ***	2.90 ***	2.77 ***	1.47 *
Leaf area index (LAI-90)	980.37 **	6.85 ***	3.16 ***	13,123.10 ***	3.87 ***	2.09 **	130.68 **	3109.88 ***	28.29 **	7.56 ***	2.92 ***	3.40 ***	1.77 **
Grain yield (GY)	10,364.12 ***	9.71 ***	5.25 ***	2767.47 ***	8.30 ***	3.36 ***	21.48 *	8794.04 ***	0.28 ^ns^	16.23 ***	1.65 *	7.63 ***	0.81 ^ns^
Harvest index (HI)	2.11 ^ns^	2.56 ***	1.70 *	27.83 *	5.46 ***	1.67 *	2.36 ^ns^	14.47 *	1.58 ^ns^	5.62 ***	1.48 *	2.81 ***	0.58 ^ns^
Relative growth rate (RGR)	104.73 **	7.60 ***	3.90 ***	375.69 **	4.53 ***	2.07 **	234.51 **	291.81 ***	2.91 ^ns^	9.84 ***	2.11 ***	4.25 ***	1.61 *
Net assimilation rate (NAR)	203.95 **	6.66 ***	3.82 ***	1136.09 ***	4.74 ***	3.45 ***	22.07 *	642.28 ***	1.82 ^ns^	9.46 ***	1.95 **	5.30 ***	1.98 **
Leaf area duration (LAD)	2149.09 ***	8.57 ***	2.93 ***	237,457.21 ***	2.96 ***	1.47 ^ns^	685.05 **	7846.93 ***	14.30 *	8.23 ***	2.73 ***	2.95 ***	1.30 ^ns^
Leaf area ratio (LAR)	534.74 **	1.33 ^ns^	1.41 ^ns^	533.14 **	1.61 *	2.40 ***	1009.90 ***	1066.07 ***	1.68 ^ns^	2.23 ***	0.72 ^ns^	2.16 ***	1.68 *
Specific leaf area (SLA)	1753.83 ***	2.13 **	1.89 **	866.78 **	1.74 *	1.45 ^ns^	87.53 *	2497.53 ***	41.95 **	3.23 ***	0.66 ^ns^	2.68 ***	0.67 ^ns^
Specific leaf weight (SLW)	1072.18 ***	3.25 ***	2.77 ***	584.00 **	2.06 **	1.81 **	75.43 *	1633.05 ***	53.51 **	4.52 ***	1.03 ^ns^	3.82 ***	0.95 ^ns^

* *p* ≤ 0.05, ** *p* ≤ 0.01, *** *p* ≤ 0.001, ns—not significant.

**Figure 1 plants-13-00882-f001:**
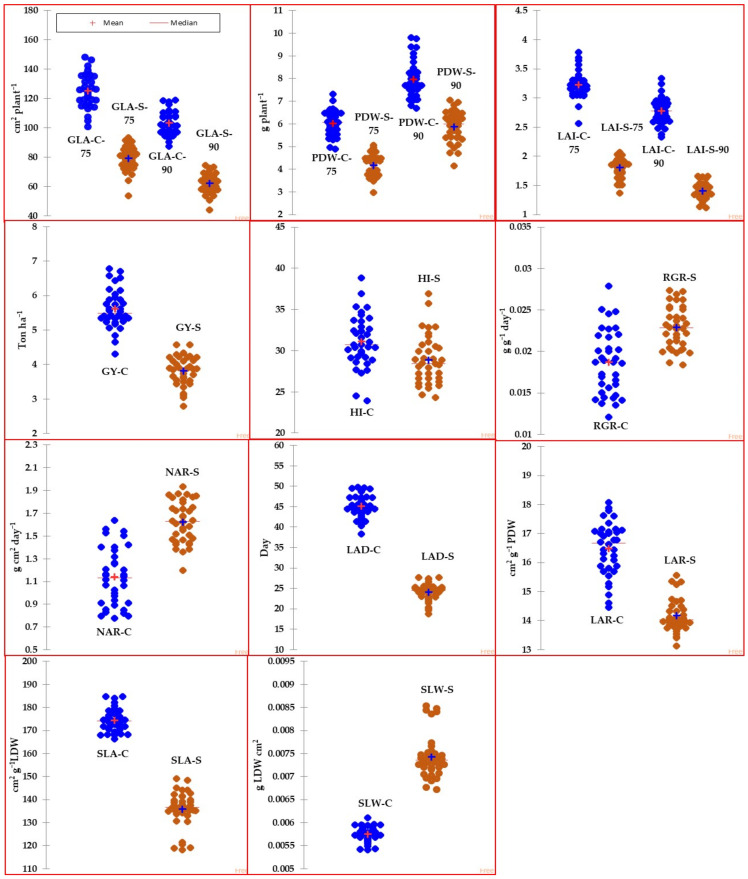
Box plots representing the variability of different agro-morphological traits measured at 75 and 90 days after sowing along with growth indicators for 36 genotypes grown under normal (C) and salinity (S) conditions. Data are the mean of pool data for two years. The full names of traits and GIs are mentioned in Table 1.

**Table 2 plants-13-00882-t002:** Descriptive statistics of different agro-morphological traits measured at 75 and 90 days after sowing along with growth indicators (GIs) among 36 genotypes under normal and salinity conditions. Data are the mean of pool data for two years. The full names of traits and GIs are mentioned in Table 1. The units of traits and GIs are mentioned in Figure 1.

Statistic	GLA-75	GLA-90	PDW-75	PDW-90	LAI-75	LAI-90	GY	HI	RGR	NAR	LAD	LAR	SLA	SLW
	Normal Conditions													
Minimum	100.79	87.36	4.90	6.70	2.56	2.33	4.30	23.93	0.012	0.779	38.39	14.46	166.34	0.0054
Maximum	148.32	119.10	7.32	9.82	3.79	3.34	6.78	38.82	0.028	1.638	49.80	18.08	184.83	0.0061
Range	47.53	31.74	2.43	3.12	1.22	1.01	2.48	14.89	0.016	0.859	11.41	3.62	18.49	0.0007
1st Quartile	117.72	97.43	5.59	7.45	3.08	2.61	5.31	29.19	0.015	0.913	43.55	15.86	170.58	0.0057
3rd Quartile	134.55	107.94	6.48	8.31	3.30	2.92	5.90	33.08	0.022	1.331	47.26	17.10	177.23	0.0059
Mean	125.21	103.32	6.03	7.97	3.23	2.78	5.61	31.13	0.019	1.139	45.06	16.48	174.41	0.0058
Standard error	1.88	1.32	0.09	0.14	0.04	0.04	0.09	0.53	0.001	0.042	0.46	0.15	0.83	0.0000
**Variation**	**1.5**	**1.4**	**1.5**	**1.5**	**1.5**	**1.4**	**1.6**	**1.6**	**2.3**	**2.1**	**1.3**	**1.3**	**1.1**	**1.1**
	**Salinity conditions**													
Minimum	53.81	44.19	2.98	4.16	1.37	1.11	2.79	24.29	0.018	1.198	18.76	13.13	118.10	0.0067
Maximum	93.53	74.51	5.06	7.04	2.07	1.66	4.58	36.93	0.027	1.936	27.68	15.56	149.03	0.0085
Range	39.71	30.32	2.08	2.88	0.70	0.55	1.79	12.63	0.009	0.737	8.92	2.42	30.93	0.0018
1st Quartile	74.90	58.04	3.76	5.42	1.72	1.34	3.55	26.63	0.021	1.468	22.96	13.84	134.21	0.0072
3rd Quartile	85.12	65.96	4.49	6.37	1.90	1.49	4.14	30.59	0.025	1.764	25.31	14.40	140.31	0.0075
Mean	79.28	62.10	4.17	5.88	1.80	1.41	3.82	28.90	0.023	1.623	24.10	14.17	135.83	0.0074
Standard error	1.36	1.09	0.08	0.11	0.03	0.02	0.07	0.50	0.000	0.030	0.37	0.09	1.31	0.0001
**Variation**	**1.7**	**1.7**	**1.7**	**1.7**	**1.5**	**1.5**	**1.6**	**1.5**	**1.5**	**1.6**	**1.5**	**1.2**	**1.3**	**1.3**
**Reduction (%)**	**36.7**	**39.9**	**30.8**	**26.2**	**44.1**	**49.3**	**31.9**	**7.1**	**−22.6**	**−42.5**	**46.5**	**14.0**	**22.1**	**−29.1**

Reduction (%) refers to the percentage decrease in the values of traits or GIs under salinity conditions compared to control conditions.

### 2.3. Correlational Studies under Normal and Salinity Conditions

The degree of association between all variables (traits and GIs) under both conditions was assessed using a Pearson correlation matrix, as shown in Table 3. Overall, the salinity stress conditions showed a stronger correlation between variables compared to the normal conditions. Additionally, all variables displayed positive correlations with each other, except for LAR and SLW, which had negative correlations with the other variables (Table 3). Morphological traits (GLA, PDW, and LAI) showed a strong positive correlation with each other (r ranged from 0.78 to 0.95) and a moderate to strong correlation with GY (r ranged from 0.44 to 0.70) under salinity conditions. Meanwhile, under normal conditions, these traits exhibited a moderate to strong correlation with each other (r ranged from 0.31 to 0.86) and a weak to moderate correlation with GY (r ranged from 0.15 to 0.49; Table 3). Three GIs related to morphological components (LAD, SLA, and SLW) exhibited a strong correlation with morphological traits and GY under salinity conditions (r ranged from 0.63 to 0.97), while they showed a weak to strong correlation under normal conditions (r ranged from 0.25 to 0.81). Under salinity conditions, GIs related to physiological components (RGR and NAR) exhibited a moderate correlation with PDW-90, LAI-75, and LAI-90 (r = 0.34–0.45), and a strong correlation with GY (r = 0.84–0.89). Under normal conditions, they showed a moderate correlation with only PDW-90 (r = 0.41–0.53) and still a strong correlation with GY (r = 0.79–0.82). LAR exhibited a moderate and negative correlation with only PDW-75 and PDW-90 under both conditions (r = 0.38–0.54). HI showed a moderate correlation with only GY, SLA, and SLW under both conditions, as well as with RGR under salinity conditions (Table 3).

### 2.4. Principal Component and Hierarchical Clustering Analyses

We conducted principal component analysis (PCA) to investigate the relationship between variables and genotypes, as well as the interconnections among the variables under normal and salinity conditions (Figure 2). The results showed that the first five principal components (PCs) accounted for 92.1% and 96.9% of the overall variation in variables and genotypes under normal and salinity conditions, respectively (Figure 2). The first PC accounted for the highest percentage of the total variability among variables, explaining 44.4% and 62.8% of the total variability under normal and salinity conditions, respectively. It was significantly associated with three morphological traits (GLA, PDW, and LAI) measured at 75 and 90 DAS, as well as GY and three GIs related to morphological components (LAD, SLA, and SLW) under both conditions. Additionally, it was associated with GIs related to physiological components (RGR and NAR) but only under salinity conditions. The second PC accounted for 22.1% and 17.2% of the total variability under normal and salinity conditions, respectively. It was primarily associated with PDW measured at 75 DAS and GIs related to physiological components (RGR, and NAR) under both conditions, as well as with LAI measured at 75 DAS, GY, and HI under normal conditions. The third, fourth, and fifth PCs explained only 11.8%, 8.1%, and 5.7% of the total variability under normal conditions and 9.1%, 5.0%, and 2.8% under salinity conditions, respectively. The third PC was strongly associated with LAR under both conditions, the fourth PC with HI under salinity conditions, and the fifth PC with LAI measured at 90 DAS under normal conditions (Figure 2).

THE PCA-biplot, based on the first two PCs, organized the variables (traits and GIs) into three main groups, with each group including the same variables under both conditions (Figure 2). The first group consisted of GY, HI, and GIs related to physiological components (RGR and NAR), and one GI related to morphological components (SLA). The second group included all morphological traits measured at 75 and 90 DAS and one GI related to morphological components (LAD). The third group included two GIs related to morphological components (SLW and LAR). The variable vectors within each group were closely aligned and formed acute angles (less than 90°), which indicated a close relationship between them. The variable vectors of the third group formed an angle of 180° with the variable vectors of the first and second groups, indicating a negative relationship between the variables of the third group and those of the first and second groups (Figure 2). Additionally, the PCA-biplot successfully separated the genotypes and distributed them across the four quadrants of the biplot, which reflected a significant variation among genotypes under both conditions (Figure 2). Interestingly, there was a clear difference between the salt-tolerant genotypes Sakha 93 and Kharchia and the salt-sensitive genotype Sakha 61. The two salt-tolerant genotypes were always located in the opposite direction to the salt-sensitive genotype under both conditions. The salt-tolerant genotypes were located in the quadrant with the highest first and second PCs, while the salt-sensitive genotype was found in the quadrant with the lowest first and second PCs under salinity conditions. Under normal conditions, the two salt-tolerant genotypes were located in the quadrant with the highest PC1 and lowest PC2, while the salt-sensitive genotype was found in the quadrant with the highest first and second PCs (Figure 2).

The hierarchical clustering heatmap (HCH) confirmed the results of the PCA analysis and effectively grouped the variables into three distinct categories in both conditions. Under both conditions, the two GIs related to physiological components (RGR and NAR), one GI related to morphological components (SLA), as well as GY and HI, formed a separate group. One GI related to morphological components (LAD) was grouped with all morphological traits measured at 75 and 90 DAS to form a separate group. Two GIs related to morphological components (SLW and SLA) were separated from other variables and formed their own group (Figure 3). Additionally, the HCH effectively separated the genotypes into three and four groups under normal and salinity conditions, respectively (Figure 3). It is noteworthy that under both conditions, the salt-tolerant genotypes Sakha 93 and Kharchia were grouped together, while the salt-sensitive genotype Sakha 61 was placed in a separate group. Furthermore, the majority of genotypes in each group did not match under both conditions. For example, the first group, which exhibited the highest values for all variables except LAR and SLW, included Sakha 61 and 6 RILs from the first crossing (2, 3, 4, 5, 7, and 9) under normal conditions. However, it included three RILs from the first crossing (3, 4, and 9) and three RILs from the second crossing (2, 5, and 15) under salinity conditions. The two salt-tolerant genotypes were located in the third group under normal conditions and in the second group under salinity conditions. Genotypes in the third group had the lowest values for all morphological traits and LAD and moderate values for other variables, while the genotypes in the second group showed moderate values for all variables (Figure 3 and Table 4). The genotypes in the fourth group under salinity conditions exhibited the lowest values for all variables except SLW. This group consisted of Sakha 61, three RILs from the first crossing (5, 11, and 13), and one RIL from the second crossing (14). However, under normal conditions, Sakha 61 and RIL-5 exhibited the highest values, RIL-11 and RIL-14 exhibited the lowest values, and RIL-13 exhibited moderate values for most variables (Figure 3 and Table 4). Therefore, the genotypes and RILs in groups 1, 2, and 3 under normal conditions could be categorized as high-, intermediate-, and low-performing genotypes, respectively. Meanwhile, the genotypes and RILs in groups 1, 2, 3, and 4 under saline conditions could be classified as salt-tolerant, moderately salt-tolerant, moderately salt-sensitive, and salt-sensitive genotypes, respectively (Table 4).

**Table 4 plants-13-00882-t004:** Mean values of different traits measured at 75 and 90 days after sowing along with growth indicators (GIs) for different groups of genotypes under normal and salinity conditions. The full names of traits and GIs are mentioned in Table 1.

	GLA-75	GLA-90	PDW-75	PDW-90	LAI-75	LAI-90	GY	HI	RGR	NAR	LAD	LAR	SLA	SLW
	Normal Conditions													
	First Year													
Group-1 (7)	125.82	108.86	6.72	9.17	3.29	2.95	6.17	32.88	0.021	1.399	46.84	14.89	180.78	0.0055
Group-2 (14)	123.42	105.60	6.43	7.86	3.15	2.78	5.01	29.42	0.013	0.833	44.52	16.08	171.29	0.0059
Group-3 (15)	113.31	96.14	5.78	7.59	2.89	2.57	5.54	31.46	0.018	1.162	40.98	15.83	172.16	0.0058
	Second year													
Group-1 (7)	139.40	115.05	6.62	9.45	3.55	3.01	6.34	30.55	0.024	1.498	49.23	16.04	180.73	0.0055
Group-2 (14)	127.35	106.30	5.99	7.75	3.38	2.84	5.35	30.36	0.017	1.015	46.65	17.08	174.11	0.0058
Group-3 (15)	129.86	97.54	5.35	7.42	3.31	2.74	5.88	32.56	0.022	1.225	45.38	17.89	173.93	0.0058
**Groups**	**Salinity conditions**													
	First year													
Group-1 (6)	87.09	71.27	4.83	6.86	1.86	1.46	3.99	30.13	0.024	1.725	24.89	13.66	137.27	0.0073
Group-2 (15)	82.41	61.07	4.57	6.25	1.75	1.28	3.70	28.58	0.021	1.583	22.73	13.30	132.93	0.0076
Group-3 (10)	72.20	54.75	3.74	5.27	1.61	1.16	3.75	31.37	0.023	1.630	20.76	14.14	130.54	0.0077
Group-4 (5)	58.34	43.38	3.46	4.50	1.22	0.95	2.93	27.44	0.017	1.371	16.29	12.78	110.82	0.0091
	Second year													
Group-1 (6)	89.59	71.89	4.51	6.64	2.11	1.79	4.51	29.35	0.026	1.776	29.27	14.60	145.70	0.0069
Group-2 (15)	83.62	67.09	4.29	6.14	1.98	1.61	4.03	28.06	0.024	1.659	26.94	14.56	143.89	0.0070
Group-3 (10)	78.11	62.63	3.90	5.67	1.89	1.54	3.94	29.32	0.025	1.697	25.75	14.86	142.84	0.0070
Group-4 (5)	72.53	59.81	3.56	4.94	1.77	1.40	3.28	26.09	0.022	1.417	23.80	15.70	128.38	0.0078

**Figure 3 plants-13-00882-f003:**
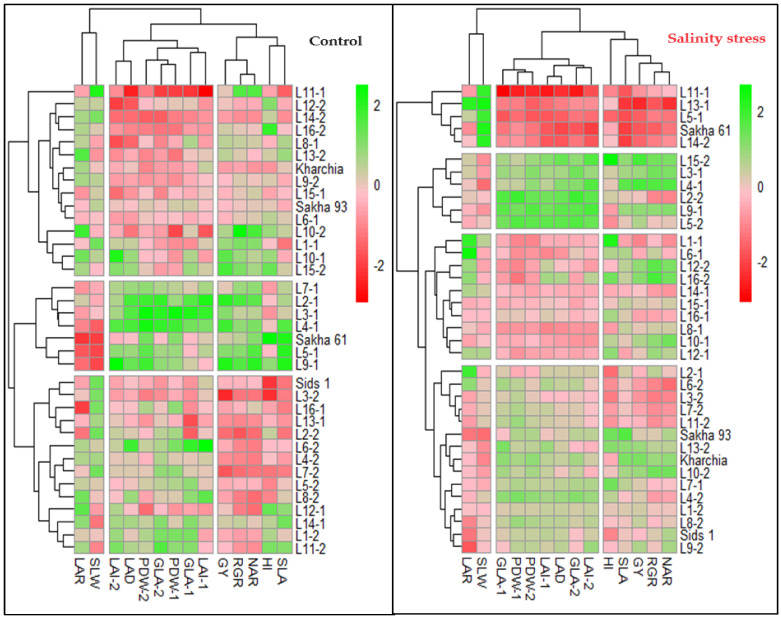
Hierarchical clustering heatmap displaying the relationships between the 36 genotypes and all variables under both normal and salinity conditions. Columns represent the variables and rows represent the genotypes. The different colors and their intensities represent the degree associations between variables and genotypes. Darker green and darker red indicate higher and lower values of the mean valuables, respectively.

### 2.5. Relationship between Growth Indicators and Grain Yield for Each Group under Normal and Salinity Conditions

Figure 4 represents the relationship between different GIs and GY under normal and salinity conditions. Under normal conditions, the two GIs related to physiological components (RGR and NAR) and one GI related to morphological components (LAD) showed a moderate and quadratic relationship (R^2^ = 0.42–0.59) with the GY of both high- and low-performing genotype groups. RGR exhibited a moderate and linear relationship with the GY of the intermediate-performing genotype group (R^2^ = 0.44), while NAR and LAD did not show a significant relationship with the GY of this group (Figure 4). The other three GIs related to morphological components (LAR, SLA, and SLW) did not show a significant relationship with the GY of both high- and low-performing genotype groups. However, they did show a weak to moderate relationship with the GY of the intermediate-performing genotype group (R^2^ = 0.31–0.54) and this relationship was linear for SLA and SLW, and quadratic for LAR (Figure 4).

Under salinity conditions, the two GIs related to physiological components (RGR and NAR) and two GIs related to morphological components (SLA and SLW) showed a strong and linear relationship (R^2^ = 0.76–0.98) with the GY of salt-tolerant and salt-sensitive genotype groups. RGR and NAR still had a moderate to strong and linear relationship (R^2^ = 0.60–0.87) with the GY of intermediate salt-tolerant and intermediate salt-sensitive genotype groups. However, SLA and SLW had a moderate and quadratic relationship (R^2^ = 0.50) with the GY of the intermediate salt-tolerant genotype group, while they did not show a significant relationship with the GY of the intermediate salt-sensitive genotype group. The other GIs related to morphological components (LAD and LAR) showed a strong (R^2^ = 0.86) and moderate (R^2^ = 0.57) relationship, respectively, with only the GY of the salt-sensitive genotype group. However, they did not show a significant relationship with the other three groups (Figure 4).

## 3. Discussion

Developing salt-tolerant genotypes is one of the most effective options for mitigating the negative impacts of salinity stress on crop growth and production. Nevertheless, creating salt-tolerant genotypes is a challenging task. Exploring the genetic diversity among different wheat genotypes is an important step in breeding programs that aim to identify potential candidates with salt tolerance. Achieving this step requires two important things: assessing the salt tolerance of wheat genotypes in real field conditions, as well as identifying traits that can be used as reliable screening criteria for distinguishing salt tolerance among wheat genotypes. In this study, we investigated the salt-tolerance of 36 wheat germplasms, including 4 cultivars differing in their salt tolerance and 32 RILs. The salt-tolerance of these germplasms was evaluated in real field conditions using different traits and GIs as screening criteria. To our knowledge, the use of GIs as screening criteria for assessing the salt tolerance of wheat genotypes is not widely used in breeding programs. The findings of this study showed a highly genetic variability for all traits and GIs between SL and G (Table 1). Furthermore, the traits and GIs varied by approximately 1.5-fold between different genotypes under both normal and salinity stress conditions (Figure 1 and Table 2). These findings indicate that both traits and GIs could effectively classify genotypes into distinct groups under normal and salinity stress conditions in real field conditions.

Generally, genotypes can enhance their performance (growth and production) by adjusting their morphological and/or physiological components in response to environmental conditions. These adjustments can lead to improved growth and production under normal conditions, as well as better adaptation and tolerance to environmental stress. Because GIs are indices that reflect the interaction of physiological, morphological, and phenological aspects, as well as being largely affected by the balance between source and sink organs and biomass allocation [21,32,34,36,37,38,39,40], we believe that these GIs can be used as screening criteria to understand genotypic differences in growth and yield under either normal or stress conditions. For instance, RGR, which is a compound rate of change, reflects the increase of PDW per unit of biomass and per unit of time. Furthermore, this index is a product of NAR (a physiological component) and LAR (a morphological component). NAR is an index that shows the balance between the rates of photosynthesis and respiration in the whole plant. This indicates the rate at which a plant converts energy from photosynthesis into biomass. LAR is an index that measures the efficiency of plants in capturing light and converting it into biomass. A high LAR indicates that the plants have a large leaf area relative to their biomass, while a low LAR indicates the opposite. SLA is an index that measures the ratio of leaf area per unit leaf mass. A high SLA indicates the plants tend to have thinner leaves and a larger leaf area relative to their mass, while a low SLA indicates the opposite. SLW is another important index that refers to the weight of a leaf per unit leaf area. It is often related to chlorophyll content, photosynthetic function, and the impact of environmental factors on leaf structure and function. Therefore, it is often used as an indicator of plant health and stress [41]. The aforementioned facts about different GIs indicate that, by analyzing these indicators, it would be possible to explain whether genotypic variation in growth and production under normal and salinity stress conditions, as well as salt tolerance under salinity stress conditions, can be attributed to morphological and/or physiological components. In this study, we observed a strong positive correlation of GY with the two GIs related to physiological components (RGR and NAR) under both conditions. Furthermore, one GI related to morphological components (LAD) showed a strong positive correlation with the three morphological traits (GLA, PDW, and LAI) measured at 75 and 90 DAS under both conditions. However, the correlation of the three morphological traits with the other GIs related to morphological components (SLA and SLW) was tighter under salinity than under normal conditions. Under both conditions, the correlation of one GI related to morphological components (LAR) with all traits was weak and non-significant, except for that of PDW (Table 3). All of these findings indicate that (1) the genotypic variation in GY under normal and salinity conditions was strongly attributed to the variation in physiological components (RGR and NAR), but not with LAR; (2) one GI related to morphological components (LAD) played a vital role in enhancing growth and production of genotypes under both conditions; and (3) SLA and SLW, which are morphological components associated with leaf structure and function, were significantly affected by salinity stress, with SLA decreasing significantly due to salinity stress compared to under normal conditions, while the opposite was found with SLW (Figure 1 and Table 2). These findings are consistent with the results reported by Cramer et al. [36], Murillo-Amador et al. [42], and Chen et al. [43]. They found that SLA values were higher under salinity stress compared to the control, and vice versa for SLW. The low SLA under salinity stress indicates that plants develop thicker leaves and a smaller leaf area relative to their mass. This suggests that leaf expansion may play an important role in the salt tolerance of genotypes. This may be due to the different components of salinity stress (osmotic stress, ionic toxicity, and essential nutrient deficit) interacting together to limit cell division and elongation. In addition, increased leaf thickness due to decreased SLA could create a longer path for carbon dioxide diffusion from stomata to chloroplasts, leading to a significant reduction in the photosynthetic capacity of plants. The increase in SLW under salinity conditions indicates that salinity stress affected the dry weight of leaves to a greater extent than their leaf area. Meanwhile, salinity stress resulted in the allocation of less biomass and photosynthetic products in leaves due to the accumulation of toxic ions and their negative impacts on the photosynthesis rate. Therefore, the sensitivity of GIs related to morphological components, particularly LAD, SLA, and SLW, to salinity stress suggests that they could be used as screening criteria to evaluate the salt tolerance of wheat genotypes.

To understand the relationship between traits and genotypes, as well as classify traits and genotypes into distinct groups, both PCA and HCH were used to achieve this goal (Figure 2 and Figure 3). In PCA, the GIs related to physiological components (RGR and NAR) and SLA were grouped with GY under both conditions. The vectors of these GIs and GY formed acute angles between each other. This indicates that improving RGR, NAR, and SLA in wheat breeding could be recognized as an effective strategy to enhance production under normal and salinity stress conditions. Our results also indicate that LAD is an important index associated with wheat growth under both conditions. This index is closely related to the three morphological traits (GLA, PDW, and LAI) measured at 75 and 90 DAS, and their vectors also form acute angles with the vectors of morphological traits (Figure 2). The high LAD refers to the higher photosynthetic capacity of leaves and, therefore, the amount of biomass produced by the plant [44]. Generally, salinity stress leads to modifications in plant morphological features such as LAD [3], which results in a reduction in the rate and duration of photosynthetic efficiency and consequently the amount of dry matter produced by photosynthesis. This may explain why the three morphological traits are closely associated with LAD under both conditions. This result also supports that LAD could be used as an effective screening criterion for evaluating the salt tolerance of wheat genotypes at early growth stages. The results also found that the vectors of SLA formed a straight angle with GY under both conditions. This confirms that improving SLW plays a vital role in enhancing crop production, whether crops are grown under normal or stressful conditions. Chen et al. [43] reported that an increase in SLW under salinity stress reflects the ability of plants to allocate more organic matter to RuBP carboxylase and photosynthesis-related thylakoids. This ultimately leads to a higher capacity of plants for photosynthetic rate and the accumulation of photosynthetic products in leaves. Therefore, we confirm that SLW could be related to enhanced leaf photosynthesis and yield potential of the wheat crop, which suggests that this indicator could be used as a screening criterion for evaluating genotypes under either normal or salinity stress conditions. Several studies have reported that SLW is highly correlated with photosynthetic capacity and dry matter accumulation under both normal and stress conditions, as well as under different agronomic practices. This ultimately contributes favorably to final GY [34,42,45,46,47].

One important finding of this study is that the different GIs did not classify the genotypes into similar groups under both normal and salinity stress conditions. For instance, the group that included the salt-sensitive genotype Sakha 61 had the lowest values for LAR and SLW and the highest values for other GIs under normal conditions. However, the group that included this genotype under salinity conditions showed opposite results compared to under the control conditions. The same situation was observed also with the salt-tolerant genotypes Sakha 93 and Kharchia (Figure 3 and Table 4). This finding suggests that when using GIs as screening criteria for grouping genotypes, it is necessary to group genotypes separately for each growth condition. This is because the growth behaviors of genotypes may vary under different environmental conditions. Therefore, we recommend using different GIs to explain the variation in growth and yield potential among genotypes under normal conditions, as well as to explain the variation in salt tolerance among genotypes under salinity stress. This is because the different GIs are efficient in detecting changes that occur simultaneously in morphological and physiological components at a whole-plant level. Therefore, this helps to determine whether variations among genotypes are due to morphological and/or physiological changes, which mainly depend on the behavior of genotypes under specific growth conditions. In this study, when the GIs were related to GY for different genotype groups under each condition, we found that, under normal conditions, GIs related to physiological components (RGR and NAR) and LAD showed a significant relationship with the GY of both high- and low-performing genotype groups. Meanwhile, GIs related to morphological components (SLA and SLW) did not show a significant relationship with the GY of both groups. Under salinity conditions, all GIs related to both morphological and physiological components showed a strong relationship with the GY of salt-tolerant and salt-sensitive genotype groups (Figure 4). These findings indicate that physiological components are the main factor explaining variations in yield performance among genotypes under normal conditions, while both physiological and morphological components play a major role in explaining salt tolerance variations among genotypes under salinity stress conditions. The studies of Praxedes et al. [21], Bayuelo-Jimenez et al. [22], and Simane et al. [48] also found that the differences in stress tolerance (salinity or drought) among genotypes were due to both physiological and morphological components of GIs. An explanation for this finding is that the different components of salinity stress (osmotic stress, ion toxicity, and nutrient imbalance) not only inhibit the physiological processes of plants but also lead to substantial changes in leaf traits such as leaf area, leaf expansion, necrosis, browning, and senescence.

## 4. Materials and Methods

### 4.1. Plant Materials and Experimental Conditions

A total of 36 wheat germplasms was used in this study and evaluated under normal and salinity stress conditions in real field conditions. These germplasms included 36 F8 recombinant inbred lines (RILs), developed by crossing the salt-sensitive genotype Sakha-61 with the salt-tolerant genotype Sakha-93 (16 RILs), and the moderately salt-tolerant genotype Sids 1 with Sakha-93 (16 RILs), as well as the salt-tolerant genotype Kharchia. The three parents and Kharchia were chosen based on previous studies that examined their salt tolerance in real field and greenhouse conditions using various agro-physiological traits [12,30,49,50]. All genotypes were evaluated under normal and high salinity concentration (150 mM NaCl) at the Experimental Farm Station of the College of Food and Agriculture Sciences at King Saud University in Riyadh, Saudi Arabia (24°25′N, 46°34′E, 400 m a.s.l.) during the winter seasons of 2019–2020 and 2020–2021. The soil of the experimental site was characterized as a sandy loam (14.9% clay, 28.4% silt, and 56.7% sand) with a bulk density of 1.48 g cm^−3^, organic matter of 0.45%, pH range of 7.80–7.90, and electrical conductivity (EC) of 1.12 dS m^–1^. The winter season at the experimental site is often sunny, with precipitation, relative humidity, and mean temperature of 8.0–25.0 mm, 17.7–47.5%, and 12.9−32.2 °C, respectively.

### 4.2. Experimental Design, Agronomic Practices, and Salinity Treatments

The field trial was carried out in a split-plot design with three replications, with the salinity treatments arranged in the main plot and the genotypes in the subplots. The size of the subplot was 1.5 m^2^ (five 1.5 m long rows spaced 20 cm apart). The seeds of the genotypes were sown by hand on 25 November in the first growing season, and 17 November in the second growing season at a seeding rate of 15 g m^−2^. The plants of wheat were fertilized with 150 kg of N, 100 kg of P_2_O_5_, and 90 kg of K_2_O per hectare. The nitrogen (N), phosphorus (P), and potassium (K) fertilizers were applied in the form of ammonium nitrate (33.5% N), calcium superphosphate (18.5% P_2_O_5_), and potassium chloride (50% K_2_O), respectively. The entire amount of P was applied before sowing. The amounts of N and K were divided into three and two equal doses, respectively. The first dose of each was applied before sowing; the second dose of N was applied at the late tillering growth stage; and the last dose of each was applied at the booting growth stage.

All genotypes in the normal treatment were irrigated with fresh water throughout the growth cycle, whereas in the salinity treatment, fresh water was used for the first 21 days after sowing in order to obtain a high germination percentage and good seedling establishment. Afterward, saline water containing 8.8 g NaCl L^−1^ (150 mM NaCl) was used for irrigation until the last irrigation. The fresh and saline water were applied through a low-pressure surface irrigation system, which consisted of a mainline (76 mm in diameter) connected to a five-cubic-meter plastic water tank and branched off at each subplot to the sub-main hoses. To control the amount of water delivered to each subplot, each sub-main hose was equipped with a manual control valve. The irrigation rate and frequency were determined according to wheat plant phenology and climatic conditions at the experimental site. In the salinity treatment, the build-up of salt in the root zone was monitored throughout the growth cycle by collecting soil samples from various locations at depths of 0–60 cm, and recording their EC. Based on the analysis of the different soil samples, the mean EC value did not exceed 16.3 dS m^−1^.

### 4.3. Measurements of Various Traits

At 75 and 90 days after sowing, which are the booting and anthesis growth stages in wheat, respectively, 10 plants from each subplot in both the normal and salinity treatments were manually pulled, packed in plastic bags, and brought to the laboratory to measure various growth traits. The green leaves of the 10 plants were separated and their area was measured using an area meter (LI 3100; LI-COR Inc., Lincoln, NE, USA) to determine green leaf area (GLA). The leaves and shoots of the 10 plants were oven-dried at 80 °C until their weight became constant and then weighed to record the dry weight of the green leaves (LDW) and shoots (SDW). The LDW and SDW were summed to record the final plant dry weight (PDW). GLA and ground area per plant were used to calculate leaf area index (LAI) according to Yue et al. [51]

Upon plants reaching the phase of full maturity (at approximately 150 days from sowing), two inner rows of 1.25 m long (totaling 0.75 m^2^ in area) were harvested by hand from each subplot. The plants were then air-dried for 7 days and weighed to record the biological yield (BY). The grains were then collected, cleaned, adjusted to moisture content of 14.0%, and weighed to record the grain yield (GY). Both BY and GY were finally expressed as tons per hectare and used to calculate the harvest index (HI), which is the ratio of GY to BY.

### 4.4. Calculation of Different Growth Indicators

Based on the values of PDW (W1 and W2), GLA (GLA1 and GLA2), and GLDW (GLDW1 and GLDW2) measured at 75 (T1) and 90 (T2) days after sowing, various growth indicators were calculated according to Hunt [52]. The abbreviation, full name, and formula of each growth indicator are presented in Table 5.

### 4.5. Data Analysis

Before statistical analysis, the normality distribution and homogeneity of variances of all traits were tested using Shapiro–Wilk and Bartlett’s chi-square tests, respectively. Sub-sequently, the data of various traits were subjected to analysis of variance (ANOVA) ap-propriate to a split-plot design to detect the effects of the main factors (salinity and genotypes) and their interaction on these traits: ANOVA was carried out for each year separately and combined over two years. The analysis of the level of correlation among traits and GIs of each other was performed using the non-parametric Spearman test. This analysis was performed using RStudio 1.3.959 (RStudio Team 2020). All traits and genotypes were subjected to PCA to detect the interrelationships among various variables and reduce the dimensionality of the dataset without losing important information. Eigenvalues, cumulative variability, and a PCA-biplot constructed between the first two components (PC1 and PC2) were extracted using the XLSTAT package version 2021. Hierarchical clustering heatmap analysis was performed for different GIs to group genotypes into distinct groups under normal and salinity conditions. Relationship analyses between different GIs and GY of each group under normal and salinity conditions were performed using GIs as independent variables, while GY was the dependent variable. Figures of regression were derived using the Sigma Plot 14.0 software program (Systat software, Inc., Chicago, IL, USA).

## 5. Conclusions

From the results of the different analyses applied in this study, we concluded that significant differences in various traits and GIs were observed between SL, G, and their interaction in each growing season and the combined data of the two seasons. The correlation between traits and GIs was higher under salinity stress than under normal conditions. PCA and HCA organized the traits and GIs into three main groups, with each group including the same variables under both conditions. Traits and GIs successfully categorized the genotypes into different groups, but did not group the genotypes similarly under both normal and salinity stress conditions. Based on the relationship between the GY of different groups and GIs, the physiological components (RGR and NAR) were the main factors explaining variations in yield performance among genotypes under normal conditions, while both physiological and morphological components (all GIs) were the major factors explaining variations in salt tolerance among genotypes under salinity stress conditions. Collectively, our findings in this study indicate that different GIs are effective as screening criteria for understanding and selecting genotypes with good performance under normal conditions and their salt tolerance under salinity stress conditions.

## Figures and Tables

**Figure 2 plants-13-00882-f002:**
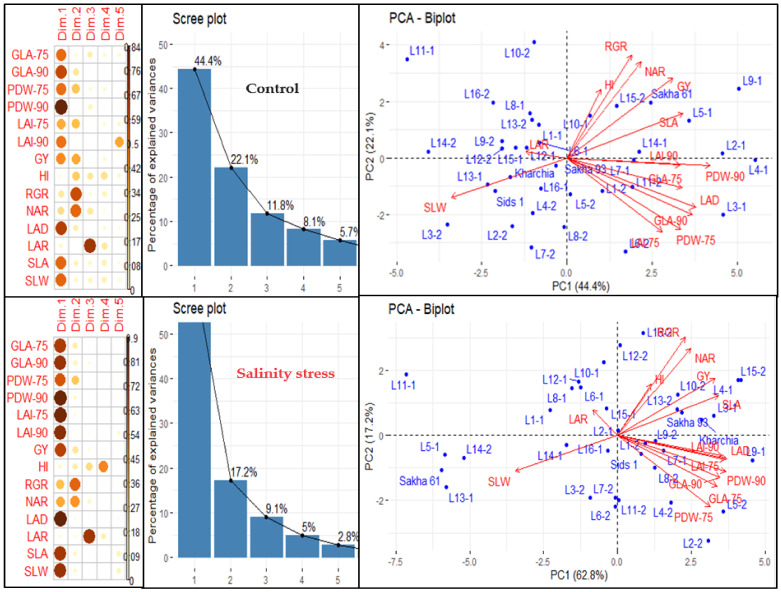
Principal component analysis (PCA) of different traits and growth indicators (GIs) in 36 genotypes under normal and salinity conditions. PCA biplot illustrates the grouping of traits and GIs in the first two principal components (PCs). Scree plot shows the percentage of explained variances of each PC, as well as the contribution of traits and GIs on extracted PCs based on square cosine and squared coordinates. The full names of the traits and GIs are mentioned in Table 1.

**Figure 4 plants-13-00882-f004:**
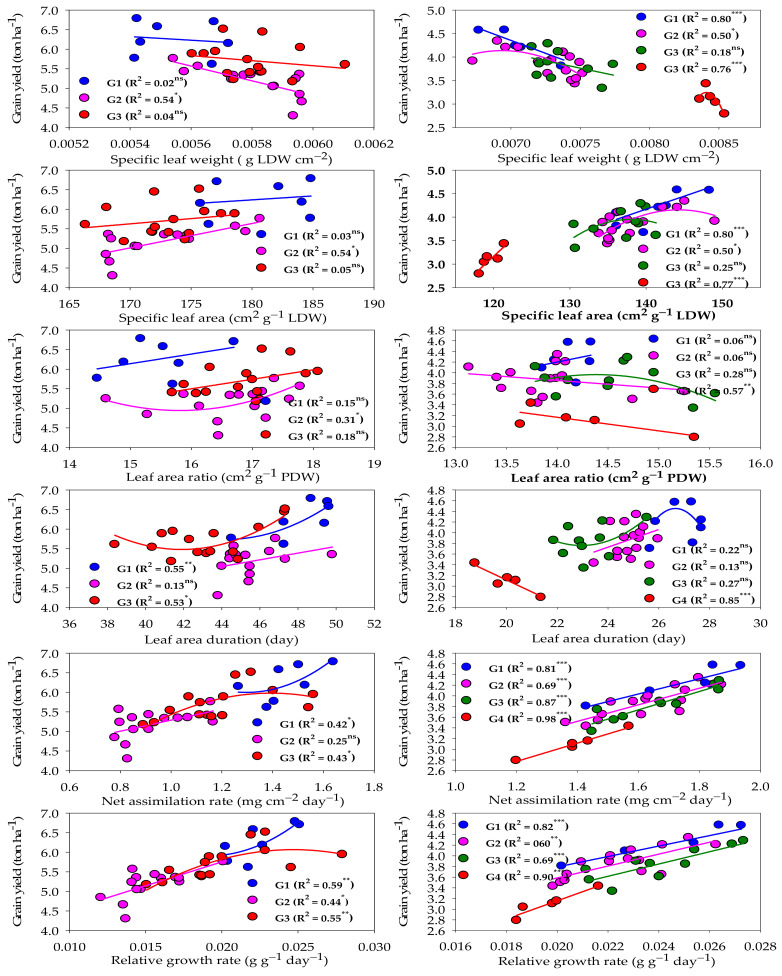
Fit regression models of the relationship between different growth indicators and grain yield for each genotype group (G) clustered under normal (left side) and salinity (right side) conditions. *** Significant at *p* < 0.0001; ** significant at *p* < 0.001; * significant at *p* < 0.05; ns: not significant.

**Table 3 plants-13-00882-t003:** Correlation matrix between all variables under normal (upper right) and salinity stress (lower left) conditions across two years. The full names of traits and GIs are mentioned in Table 1. The units of traits and GIs are mentioned in Figure 1.

Variables	GLA-75	GLA-90	PDW-75	PDW-90	LAI-75	LAI-90	GY	HI	RGR	NAR	LAD	LAR	SLA	SLW
**GLA-75**		0.68 ***	0.58 ***	0.62 ***	0.63 ***	0.43 **	0.42 **	0.08	0.16	0.09	0.65 ***	−0.25	0.48 ***	−0.49 ***
**GLA-90**	0.91 ***		0.86 ***	0.78 ***	0.61 ***	0.65 ***	0.24	0.00	0.02	0.03	0.78 ***	−0.18	0.43 ***	−0.44 ***
**PDW-75**	0.92 ***	0.88 ***		0.83 ***	0.66 ***	0.46 ***	0.09	0.09	0.17	0.02	0.69 ***	−0.51 ***	0.34 *	−0.36 *
**PDW-90**	0.93 ***	0.92 ***	0.95 ***		0.54 ***	0.57 ***	0.55 ***	0.06	0.41 **	0.53 ***	0.69 ***	−0.54 ***	0.56 ***	−0.56 ***
**LAI-75**	0.90 ***	0.87 ***	0.82 ***	0.90 ***		0.31	0.15	0.27	0.13	0.10	0.81 ***	−0.09	0.25	−0.28
**LAI-90**	0.78 ***	0.90 ***	0.77 ***	0.85 ***	0.87 ***		0.49 ***	0.18	0.25	0.27	0.81 ***	−0.12	0.40 *	−0.40 *
**GY**	0.53 ***	0.57 ***	0.44 ***	0.68 ***	0.70 ***	0.68 ***		0.42 **	0.82 ***	0.79 ***	0.39 **	−0.03	0.60 ***	−0.58 ***
**HI**	0.15	0.16	0.02	0.13	0.21	0.25	0.33 *		0.25	0.21	0.05	−0.12	0.47 ***	−0.47 ***
**RGR**	0.21	0.29	0.02	0.34 *	0.41 **	0.41 **	0.84 ***	0.35 *		0.96 ***	0.07	−0.10	0.40 *	−0.37 *
**NAR**	0.24	0.29	0.16	0.45 ***	0.44 **	0.44 **	0.89 ***	0.28	0.94 ***		0.10	−0.26	0.43 **	−0.39 **
**LAD**	0.87 ***	0.91 ***	0.82 ***	0.91 ***	0.97 ***	0.96 ***	0.71 ***	0.24	0.42 **	0.45 ***		−0.13	0.40 *	−0.42 **
**LAR**	−0.13	−0.05	−0.42**	−0.38 *	−0.15	−0.13	−0.24	0.20	0.08	−0.25	−0.14		−0.08	0.06
**SLA**	0.63 ***	0.67 ***	0.63 ***	0.72 ***	0.72 ***	0.69 ***	0.84 ***	0.34 *	0.71 ***	0.72 ***	0.73 ***	−0.11		−0.99 ***
**SLW**	−0.66 ***	−0.69 ***	−0.56 ***	−0.74 ***	−0.75 ***	−0.70 ***	−0.83 ***	−0.33 *	−0.69 ***	−0.71 ***	−0.75 ***	0.11	−1.00***	

* *p* ≤ 0.05, ** *p* ≤ 0.01, *** *p* ≤ 0.001.

**Table 5 plants-13-00882-t005:** The abbreviation (Abb.), full name, and formula of various growth indicators used in this study.

Abb.	Index Full Name	Index Formula
RGR	Relative growth rate (g g^−1^ day^−1^)	$\frac{LnW_{2}-LnW_{1}}{T_{2}-T_{2}}$
NAR	Net assimilation rate (g cm^2^ day^−1^)	$\frac{\left(W_{2}-W_{1}\right)\left(Ln{GLA}_{2}-Ln{GLA}_{1}\right)}{\left({GLA}_{2}-{GLA}_{1}\right)\left(T_{2}-T_{1}\right)}$
LAD	Leaf area duration (day)	$\frac{\left({GLA}_{1}+{GLA}_{2}\right)\left(T_{2}-T_{1}\right)}{2}$
LAR	Leaf area ratio (cm^2^ g^−1^ PDW)	$\frac{\left({GLA}_{2}-{GLA}_{1}\right)\left(LnW_{2}-LnW_{1}\right)}{\left(Ln{GLA}_{2}-Ln{GLA}_{1}\right)\left(W_{2}-W_{1}\right)}$
SLA	Specific leaf area (cm^2^ g^−1^ LDW)	$\frac{\left({GLA}_{2}-{GLA}_{1}\right)\left(Ln{GLDW}_{2}-Ln{GLDW}_{1}\right)}{\left(Ln{GLA}_{2}-Ln{GLA}_{1}\right)\left({GLDW}_{2}-{GLDW}_{1}\right)}$
SLW	Specific leaf weight (g LDW cm^2^)	$\frac{\left(W_{2}-W_{1}\right)\left(Ln{GLA}_{2}-Ln{GLA}_{1}\right)}{\left({GLA}_{2}-{GLA}_{1}\right)\left(LnW_{2}-LnW_{1}\right)}$

PDW and LDW indicate plant dry weight and leaf dry weight, respectively.

## Data Availability

All data are presented in article.

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
