# Peer review of "Ability of Different Growth Indicators to Detect Salt Tolerance of Advanced Spring Wheat Lines Grown in Real Field Conditions"

_plants, 2024, doi:10.3390/plants13060882_

Round 1
Reviewer 1 Report
Comments and Suggestions for Authors
In this study, authors analyzed several growth indicators of 36 recombinant inbred lines (RILs) along with four genotypes under normal and 150 mM NaCl in a two-year field trial. They also assessed the correlations between yield and these growth indicators trying to find which can be serve as valuable selection criteria for evaluating the performance of genotypes under normal conditions and their salt tolerance under salinity conditions. Although authors stated that different GIs are effective as screening criteria for understanding and selecting genotypes with good performance under normal conditions and their salt tolerance under salinity stress conditions, I do not think they obtained a very clear and confidential conclusion. Besides, growth parameters can be significantly influenced by environmental conditions, I doubt whether the results of two-year experiment are confidential. Also, this study is lack of novelty, there is no new ideas or new method showed in this study, which may not attract our readers.
Author Response
Reviouer#1
In this study, authors analyzed several growth indicators of 36 recombinant inbred lines (RILs) along with four genotypes under normal and 150 mM NaCl in a two-year field trial. They also assessed the correlations between yield and these growth indicators trying to find which can be serve as valuable selection criteria for evaluating the performance of genotypes under normal conditions and their salt tolerance under salinity conditions. Although authors stated that different GIs are effective as screening criteria for understanding and selecting genotypes with good performance under normal conditions and their salt tolerance under salinity stress conditions, I do not think they obtained a very clear and confidential conclusion. Besides, growth parameters can be significantly influenced by environmental conditions, I doubt whether the results of two-year experiment are confidential. Also, this study is lack of novelty, there is no new ideas or new method showed in this study, which may not attract our readers.
Response: We greatly appreciate your critical comments. We hope that our following explanation demonstrates the importance of this study and the significance of using different growth indices (GIs) as selection criteria for evaluating genotypes, especially under salinity stress conditions.
Various salt tolerance indices, such as productivity stability index, tolerance index, stress tolerance index, stress susceptibility index, average relative performance index, relative efficiency index, and relative salinity effect are commonly used to evaluate the salt tolerance of genotypes. However, these indices consider the behavior of genotypes under both control and salinity stress conditions simultaneously. This may not be effective for evaluating the genotypes whose performance differs significantly between control and salinity stress conditions. However, different growth indicators (GIs) consider the behavior of genotypes between two successive growth stages under the same conditions. Additionally, these GIs refer the relationship between the morphological and physiological behavior components of genotypes under the same conditions. This indicate that GIs are less influenced by environmental conditions when compared with salt tolerance indices. Therefore, all of these advantages of GIs making them effective for evaluating genotypes under control and salinity conditions separately and useful for selecting genotypes with good performance in growth and yield under normal conditions, as well as for discriminating the salt tolerance of genotypes under salinity stress conditions as shown in our study. To our knowledge, the use of GIs as screening criteria for assessing the salt tolerance of wheat genotypes is not yet widely used in plant breeding programs. Therefore, the significance of this study lies in its evaluation of genotypes in real field conditions and its assessment of the effectiveness of different GIs as screening criteria for evaluating the salt tolerance of genotypes.
Based on the findings of this study, our next research will investigate whether it is necessary to grow genotypes in a control treatment (without salinity) when evaluating their salt tolerance. We will compare salt tolerance indices and growth indicators as screening criteria, with the former considering the behavior of genotypes under salinity and control conditions together, and the latter focusing on their behavior under salinity and control conditions separately. If different GIs can effectively assess the salt tolerance of genotypes, it will reduce the effort and cost of evaluating the salt tolerance of a large number of genotypes, as there will be no need to assess them in control conditions.
Finally, the introduction has been revised to include cited references, the research design has been improved, and the conclusions have been supported by the results.

Reviewer 2 Report
Comments and Suggestions for Authors
The title is too long, consider shortening it.
I would title: Growth Indicators for Detecting Salt Tolerance of Advanced Spring Wheat Lines
Growth indicators do not have abilities.
The abstract should be shorter. I suggest focusing on the novelty of this experiment. We know there are different indicators of salt stress, but which is the best? Did you gain knowledge about the best growth indicator?
The manuscript has grammar
L13. of this germoplasms
L24. SLA being closely
L31. between the morphological
L39 - L105. The introduction section is very long and wordy; focus it on the problem to be addressed in this manuscript. There are too many abbreviations, which makes the manuscript difficult to read.
L41 delete "Unfortunately"
L497 which results in
Is there any difference between Figure 2 and Table 2? It seems to me thatTable 2 gives more information than the Figure.
Please highlight your results, not just present the figures and tables.
Comments on the Quality of English LanguageThe manuscript has many abbreviations which made it difficult to read and needs proofreading.
Author Response
Reviewer #2
Response: Thank you for your valuable feedback. We have made the necessary revisions to the manuscript based on your comments. We believe that these changes have significantly improved the quality of the manuscript.
- The title is too long, consider shortening it. I would title: Growth Indicators for Detecting Salt Tolerance of Advanced Spring Wheat Lines. Growth indicators do not have abilities.
Response: Thank you very much for your suggestion. In this study, we aim to assess the effectiveness of different growth indicators as screening criteria for detecting salt tolerance in wheat genotypes. Therefore, we have included "ability" in the title to reflect this focus.
- The abstract should be shorter. I suggest focusing on the novelty of this experiment. We know there are different indicators of salt stress, but which is the best? Did you gain knowledge about the best growth indicator?
Response: Thank you very much for your suggestion. The abstract has been revised to address your feedback.
- The manuscript has grammar
L13. of this germoplasms germplasm
L24. SLA being closely
L31. between the morphological
Response: Thank you very much for your comment. The germplasm in line 13 is correct word. The other two comments have been done.
- L39 - L105. The introduction section is very long and wordy; focus it on the problem to be addressed in this manuscript. There are too many abbreviations, which makes the manuscript difficult to read.
Response: Thank you very much for your suggestion. The paragraphs from line 39 until 105 have been revised to address your feedback.
- L41 delete "Unfortunately"
Response: Done.
- L497 which results in
Response: Done.
- Is there any difference between Figure 2 and Table 2? It seems to me that Table 2 gives more information than the Figure.
Response: Thank you very much for your comment. Figure 1 (box plot) illustrates the variability of traits and GIs among genotypes under normal and salinity conditions, while Table 2 provides a summary of the descriptive statistics for traits and GIs in response to both conditions across 36 genotypes.
- Please highlight your results, not just present the figures and tables.
Response: Thank you for your suggestion. The results section has been updated accordingly.
- The manuscript has many abbreviations which made it difficult to read and needs proofreading.
Response: Thank you for your feedback. We have made sure to include the full name of any abbreviations the first time they are used in the manuscript, as per the journal guidelines. Additionally, the abbreviations for growth indicators (RGR, NAR, LAD, LAR, SLA, and SLW) are widely recognized and should be clear to the readers.

Reviewer 3 Report
Comments and Suggestions for Authors
The paper „Ability of Different Growth Indicators for Detecting Salt Tolerance of Advanced Spring Wheat Lines Grown under Field Conditions” is current and very well structurate.
The authors highlighted very well the ability of different GIs for understanding and selecting genotypes with good performance under normal conditions and salt tolerance under salinity stress conditions.
I propose to publish the paper in present form.
Author Response
Reviewer #3
The paper,Ability of Different Growth Indicators for Detecting Salt Tolerance of Advanced Spring Wheat Lines Grown under Field Conditions” is current and very well structurate.
The authors highlighted very well the ability of different GIs for understanding and selecting genotypes with good performance under normal conditions and salt tolerance under salinity stress conditions.
I propose to publish the paper in present form.
Response: Thank you for your revisions to our manuscript and your support for publishing it. The different growth indicators (GIs) play a crucial role in our study, helping us understand and select genotypes with good performance under normal conditions and salt tolerance under salinity stress

Round 2
Reviewer 1 Report
Comments and Suggestions for Authors
1. Line 227, it is the Figure 1? In this figure, what is the content in the y-axis? Or what is for each panel?
2. Line 394, it is figure 3?
3. Line 401-402, line 450-475, the same figure?
Author Response
Reviewer#1
Response: Thank you for your valuable feedback. We have made the necessary revisions to the manuscript based on your comments. We believe that these changes have significantly improved the quality of the manuscript.
- Line 227, it is the Figure 1? In this figure, what is the content in the y-axis? Or what is for each panel?
Response: Thank you for pointing out the error. The box plot in question is actually labeled as Figure 1, not Figure 2. The title of y-axis has been added.
- Line 394, it is figure 3?
Response: Thank you for pointing out the error. The hierarchical clustering heatmap labeled as Figure 3, not Figure 4.
- Line 401-402, line 450-475, the same figure?
Response: Thank you for pointing out the error. The duplicate Figure 4 has been removed.
